# Current Concepts in Humeral Component Design for Anatomic and Reverse Shoulder Arthroplasty

**DOI:** 10.3390/jcm10215151

**Published:** 2021-11-02

**Authors:** Joaquin Sanchez-Sotelo

**Affiliations:** Division of Shoulder, Elbow Surgery Mayo Clinic Rochester, Rochester, MN 55905, USA; sanchezsotelo.joaquin@mayo.edu

**Keywords:** shoulder, arthroplasty, anatomic arthroplasty, reverse arthroplasty

## Abstract

The history of humeral component design has evolved from prostheses with relatively long stems and limited anatomic head options to a contemporary platform with short stems and stemless implants with shared instrumentation and the ability to provide optimal shoulder reconstruction for both anatomic and reverse configurations. Contemporary humeral components aim to preserve the bone, but they are potentially subject to malalignment. Modern components are expected to favorably load the humerus and minimize adverse bone reactions. Although there will likely continue to be further refinements in humeral component design, the next frontiers in primary shoulder arthroplasty will revolve around designing an optimal plan, including adequate soft tissue tension and providing computer-assisted tools for the accurate execution of the preoperative plan in the operating room.

The design of the humeral component in shoulder arthroplasty has evolved tremendously over the last two decades. When discussing the general principles of shoulder replacement, glenoid reconstruction is reviewed frequently. Interestingly, the humeral side of the joint is discussed less often. The purpose of this review article is to provide an update on the principles driving contemporary humeral component design.

## 1. Brief Historical Perspective

The history of humeral component design has evolved over a few important milestones that have had a major impact on where we are today.

### 1.1. From Monoblock and Cemented to Modular Cementless

Neer is considered by some to be the father of modern shoulder surgery in North America [1]. The original Neer prosthesis was a smooth monoblock hemiarthroplasty with a narrow stem and three sizes. Because the stem was narrow and designed for a cemented application, it could be “floated” in the canal in whichever location was best to position the prosthetic humeral head anatomically [2]. Although early on the original Neer prosthesis was implanted without cement, in the absence of surface treatment, cementless implantation led to a high rate of radiographic loosening [3]. The development of technology to treat the stem with ingrowth-friendly surfaces led to the successful survival of cementless humeral components [4]. At the same time, modular anatomic humeral heads were introduced to allow a humeral head size selection independent of the stem size selection [5]. Currently, most surgeons agree on trying to avoid the use of cement for humeral component fixation at the time of primary arthroplasty; If component revision becomes necessary, cement removal could substantially increase the difficulties associated with the revision procedure.

### 1.2. A More Sophisticated Understanding of Humeral Geometry

Traditional cementless ingrowth stems with standard modular heads were noted as not allowing for the anatomic restoration of the proximal humerus geometry in many shoulders: the fit of the stem in the humeral canal dictated where the head would “land”. This prompted studies on the variability of proximal humerus morphology [6] and on the design of implants with various features to adjust the position of the prosthetic head relative to the stem in terms of inclination, eccentricity, and offset [7].

### 1.3. Reverse Arthroplasty

Grammont revolutionized the field of shoulder arthroplasty with the development of the reverse prosthesis concept: a more constrained implant of reverse geometry that would increase the moment arm of the deltoid to compensate for the rotator cuff insufficiency [8]. Despite becoming an incredibly successful implant in terms of the restoration of active elevation and the long-term survivorship, [9] the limitations of the traditional Grammont-style prosthesis included poor restoration of the internal and external rotation as well as excessive impingement of the humeral polyethylene with the medial scapular pillar, leading to polyethylene wear, bone loss (notching), and eventually loosening [10]. Frankle modified the reverse principles to optimize the impingement-free range of motion and the tension of the axial rotator cuff with the design of a reverse prosthesis with a less truncated sphere and a more vertical polyethylene (135-degree opening angle) [11,12]. Contemporary designs follow the modifications of reverse introduced by Frankle.

### 1.4. Shorter Stems, Resurfacing, and Stemless

The length of most of the traditional stems was arbitrarily set to occupy the upper third to half of the humerus. Standard-length stems have demonstrated outstanding performance and survivorship [13]. However, avoiding relatively long stems is attractive for several reasons, including easier revision, easier implantation of an ipsilateral total elbow arthroplasty, and maybe easier management of periprosthetic fractures. Resurfacing arthroplasty represented a first attempt to avoid stem use, [14] but resurfacing components have fallen out of favor since incomplete head removal made glenoid access more difficult, and the prosthetic head sizes with various degrees of offset and eccentricity could not be used. As such, the design has evolved into the use of short-stem and stemless prostheses. The length of most of the short stems has been chosen arbitrarily, with few exceptions. For anatomic arthroplasty, stemmed and stemless prostheses seem to provide equivalent results, provided satisfactory implantation is achieved at the time of surgery [15,16].

### 1.5. Malalignment and Adverse Bone Reactions

One benefit of longer stems is that a tight stem fit into the endosteal canal facilitates adequate alignment. Ultrashort stems introduce two potential problems: poor alignment and adverse bone reactions.

Stems that do not engage the cylindrical portion of the endosteal canal can easily be misaligned. Excessive valgus or varus will lead to a poor humeral head position in anatomic arthroplasty. Similarly, poor alignment can lead to a reverse polyethylene that is excessively horizontal with an increased risk of notching, or to a more vertical polyethylene that may facilitate dislocation. Certain short stems have been designed with just enough length to avoid malalignment [17]. Stemless prostheses are also at risk for malalignment (Figure 1). As such, care must be taken to optimize the humeral head cut to minimize the chances of malalignment with ultrashort stem and stemless prostheses.

Certain ultrashort stems need larger diameters to achieve primary stability in the absence of diaphyseal contact. This concept has been captured with the fill–fit ratio popularized by Walch et al. [18]. Severe stress shielding with resorption of the greater tuberosity, and in extreme cases in areas of complete cortical defect, has been reported with the implantation of larger sizes of certain stems, and malalignment may accentuate these adverse bone reactions through point contact of the stem on the cortical bone, further shielding the proximal bone from stress (Figure 2). Thus, it is important to design implants that do not shield the metaphysis from stress.

### 1.6. Preoperative Planning Software and Surgical Execution

The development and widespread use of preoperative planning software has revolutionized the field of shoulder arthroplasty. I trained at a time when plain radiographs were the only imaging study obtained before shoulder arthroplasty. Today, the vast majority of shoulder arthroplasty surgeons rely on computer tomography to understand each shoulder to be replaced and to plan the surgery accordingly. Furthermore, preoperative planning software has advanced the field to a whole other level: three-dimensional renderings, automated measurements, and virtual implant overlays allow for accurate planning of the implant positioning to optimize orientation, seating, contact, motion free of impingement, and other variables [19,20,21]. Such software can then be used for artificial intelligence predictive algorithms, manufacturing patient-specific guides, and using computer-assisted surgery with navigation or robotics.

In the field of reverse shoulder arthroplasty, preoperative planning software reveals that using a larger glenoid with a larger lateral offset and an inferior overhang is the most successful strategy to optimize the range of motion free of impingement, especially when combined with a more vertical (typically 135 degree) polyethylene opening angle [22].

### 1.7. Same-Day Surgery and Ambulatory Surgery Centers

In the United States, there is a growing interest in same-day discharge after shoulder arthroplasty, as well as in performing these procedures in ambulatory surgery centers. This is driven by two main forces: the potential for certain financial gain and the need to decrease hospitalizations, especially considering the current COVID-19 pandemic [23]. Ambulatory surgery centers have less capacity to process large inventories and instrument trays. As such, there is the need for streamlined instrumentation and shared instruments between stem and stemless designs. Patient-matched implants and preoperative planning software may further help decrease inventory.

### 1.8. Proximal Humerus Bone Density

Understanding the bone mineral density of the proximal humerus is paramount to optimizing the primary stability of modern humeral components. In the osteoarthritic shoulder, the strongest bone is at the periphery and is closer to the superior aspect of the humeral head. As such, fixation is theoretically optimized by achieving a prosthetic fit to the periphery of the metaphysis and with a slightly higher humeral head cut [24]. However, one downside of performing a higher humeral head cut is the more difficult access to the glenoid.

### 1.9. Implications for Humeral Component Design

The brief historical review summarized above provides the grounds for design features that are perceived as desirable when considering contemporary humeral component design (Table 1).

## 2. Implant Configurations: What Are Our Targets on the Humeral Side Currently?

### 2.1. Anatomic Shoulder Arthroplasty

When performing an anatomic shoulder arthroplasty, the main goal on the humeral side is to restore the overall geometry of the proximal humerus. Considering the variability of the human shoulder (in terms of size, retroversion, and other parameters), as well as the need to adapt to the final position of the humeral stem or stemless nucleus, it is necessary to design systems with multiple head sizes and thicknesses as well as with a mechanism to offset the humeral head with respect to the final position of the humeral stem/nucleus (Figure 3). In most shoulders, the restoration of the premorbid anatomy provides the best outcome. However, in certain shoulders, the humeral head version, diameter, thickness, and/or eccentricity may need to be adapted to the condition of the soft tissues. For example, in a shoulder with substantial posterior subluxation and chronic stretching of the posterior rotator cuff and capsule, it may be necessary to implant a humeral head that is thicker than the premorbid native head to properly tension the soft tissue envelope posteriorly.

### 2.2. Reverse Shoulder Arthroplasty

Understanding the nuances associated with the design and implantation of the humeral component in reverse shoulder arthroplasty is not possible without considering the glenoid side [22]. Currently, most would agree that reverse shoulder arthroplasty requires a fine balance between (1) maximizing impingement-free range of motion and (2) optimizing soft tissue tension and muscle function around the shoulder.

Avoiding any impingement between the medial aspect of the polyethylene and the body of the scapula and scapular pillar essentially requires displacing the proximal humerus laterally and posteroinferiorly. This is best achieved by implanting a larger glenosphere with a posteroinferior overhang in reference to the glenoid vault combined with a larger lateral offset of the glenoid component. Larger lateral offsets may be achieved with thicker glenospheres, structural bone grating between the native glenoid and the baseplate (bio-RSA), or thicker (augmented) baseplates (Figure 4). The benefit of bio-RSA and augmented baseplates over thicker glenospheres is that both bone graft and metal augments provide adequate correction of angular deformities (inclination and retroversion) without reaming excessively, which can lead to impingement as well.

Provided the surgeon commits to implantation of large glenospheres with an inferior overhang and a lateral offset using any of the three methods above, the humeral component must allow for a minimal thickness above the cut surface in order to avoid excessive soft tissue tension secondary to lateralization, distalization, or both. A relatively easy way to design humeral components that allow for anatomic and reverse compatibility is to design humeral bearings that rest on the cut surface of the humerus, so-called onlay systems. The downside of onlay systems for those surgeons who maximize impingement-free motion on the glenoid side is that the soft tissue tension may be excessive. This can be compensated for by lowering the humeral cut, which may be acceptable in the cuff-deficient shoulder but not in the cuff-intact osteoarthritic shoulder, where a lower cut would damage the rotator cuff. As such, if the surgeon chooses to maximize impingement-free range of motion through glenoid implantation, the thinnest humeral bearing construct should place the pivot point at or below the humeral cut. When the pivot point (the deepest portion of the polyethylene) is below the humeral cut, implants are classified as inlays.

However, *the onlay vs. inlay controversy should probably be abandoned* for two reasons. Firstly, there is a high level of variability regarding how much lateralization and distalization are provided by the many implants in the market [25]. Classifying them as onlays or inlays is an oversimplification. What matters is where the humerus “lands” for a specific glenoid reconstruction, depending on the humeral implant selected and where it is implanted. This will affect the length and the moment arm of the deltoid and rotator cuffs [26]. Secondly, surgeons may implant inlay components in an onlay fashion or the other way around. For example, the original prosthesis designed by Dr. Frankle was an inlay design; however, its proximal portion was relatively large and could not be fully inset in the humeral metaphysis of many patients, thus resulting in an onlay application of an inlay design (Figure 5). By the same token, if thicker polyethylene bearings or a metal spacer are added to an inlay prosthesis to guarantee adequate stability, the pivot point is at an onlay level despite the implant being designed as an inlay. Consequently, even though implants that allow placement of the pivot point at or below the cut surface of the humerus are necessary to optimize soft tissue tension across the whole spectrum of shoulder replacements, in many shoulders, these inlay components will behave as onlay ones because thicker polyethylenes may be needed to avoid dislocation, especially in the cuff-deficient shoulder. The ideal degree of humeral lateralization probably varies from individual to individual depending on the underlying diagnosis and other characteristics.

## 3. From Design to Implantation: Pearls and Pitfalls Learned with Use of a Contemporary Humeral Component

Hopefully, a review of the history of implant component design and an understanding of what are considered contemporary targets today will help drive the surgical techniques for implantation of contemporary humeral components (Figure 6).

### 3.1. Anatomic Arthroplasty

#### 3.1.1. Preoperative Planning

Our preference is to plan the humeral head osteotomy at 2–3 mm proximal to the transition between the rotator cuff attachment and the humeral head. Some surgeons prefer performing the osteotomy at fixed angles, typically 135 degrees of inclination and 30 degrees of retroversion. Others prefer to make the cut at the exact location of the anatomic humeral neck. In such a case, it is possible that the stem will end up oriented inside the canal in varus or valgus, and the implications of malalignment in anatomic arthroplasty are less substantial, provided the humeral head is reconstructed anatomically.

#### 3.1.2. Humeral Osteotomy

Many surgeons are used to performing the humeral head osteotomy with so-called freehand techniques. However, for those implants with a fixed neck–shaft angle, it may be advantageous to use an extramedullary or an intramedullary guide. Our preference is to use an intramedullary guide, and the selection of the entry point of the guide is paramount to avoid a varus or valgus cut. A C–guide may then be used to select the ideal cut height (Figure 7).

#### 3.1.3. Sizing

The preparation of the metaphysis for modern components that rely on peripheral fixation typically aims to place the component “bowl” so that it will leave 2–4 mm of cancellous bone between the component and the cortical rim of the metaphysis (Figure 8). A wider distance may be advantageous in patients with a stronger bone that does not require maximizing peripheral fixation. The guide pin for the reaming of the metaphysis may be centered using a trial humeral head or sizing discs.

#### 3.1.4. Humeral Preparation and Implantation

Since short and stemless humeral components do not provide selfaligning features, surgeons must be extremely careful at the time of compactor preparation and component implantation to replicate the desired alignment based on the preoperative planning and the osteotomy performed. The most common pitfall is to place the component in an excessive varus. As such, an effort must be made to use the compactor/inserter handle when pushing into the varus.

#### 3.1.5. Humeral Head Selection

The resected humeral head provides a great reference for the selection of the correct diameter and also for the thickness. The geometry of the humeral head can be perfectly replicated by selecting the right combination of diameter, thickness, and eccentricity. In shoulders with a severe preoperative soft tissue imbalance, changes in the humeral head thickness or in diameter may be needed. As mentioned previously, arthritic shoulders with a severe posterior subluxation may require the use of a thicker humeral head to properly tension the posterior capsule and cuff. In shoulders with avascular necrosis, it may be wise to downsize the humeral head since there is a higher risk of stiffness. Intraoperative testing may be used to confirm an adequate soft tissue balance in anatomic shoulder arthroplasty (Table 2).

### 3.2. Reverse Arthroplasty

As mentioned before, humeral and glenoid planning are intimately related in reverse arthroplasty. The configuration and placement of the glenoid component have a major impact on the range of motion free of impingement. Humeral planning is then completed to select the correct size and alignment of the humeral component. My preference is to select a polyethylene opening angle of 135 degrees. The combined configuration of the glenoid and humeral components will lead to specific arcs of motion free of impingement. It will also lead to a specific position of the humerus in space in reference to the scapula, which will impact soft tissue tension. Currently, there is no consensus regarding the ideal position of the humerus in reference to the scapula in reverse arthroplasty, but most aim to replicate the anatomic position of the greater tuberosity from lateral to medial.

An accurate humeral cut and a correct implantation of the humeral component at the time of surgery are important to replicate the polyethylene opening angle desired for a given shoulder. The same considerations described for anatomic shoulder arthroplasty regarding humeral osteotomy and sizing, as well as humeral preparation and implantation, apply to most platform stems. However, reverse arthroplasty is more constrained than anatomic arthroplasty, and achieving primary stability of the humeral component is maybe more important. As such, we have a low threshold to implant the so-called “plus sizes”, which are slightly oversized in reference to the standard sizes to provide a tighter fit.

Regarding the bearing selection, the thinnest polyethylene will result in a pivot point at the level of the humeral cut. Thicker bearings with or without the addition of a metal tray will move the pivot point proximal and medial with reference to the geometric center of the proximal humerus, and as such will increase humeral lateralization and distalization. The ideal bearing thickness is typically selected based on intraoperative trialing, and currently there are no good objective parameters to guide the bearing selection. Bearings with improved wear performance, such as vitamin E polyethylene, are definitively attractive.

## 4. Future Directions

The evolution of humeral component design has been quite remarkable. Contemporary implants provide the opportunity for bone preservation, platform convertibility, the anatomic reconstruction of the proximal humerus when anatomic arthroplasty is performed, and optimal arcs of motion free of impingement with adequate soft tissue tension when reverse arthroplasty is performed. However, the jury is still out regarding the potential for component malalignment and bone adaptation to these newer components over time. The preoperative planning software is very refined, but the execution of the plan is still evolving. Various navigation and robotic systems are being developed and will likely translate into a more accurate execution of the preoperative plans.

## Figures and Tables

**Figure 1 jcm-10-05151-f001:**
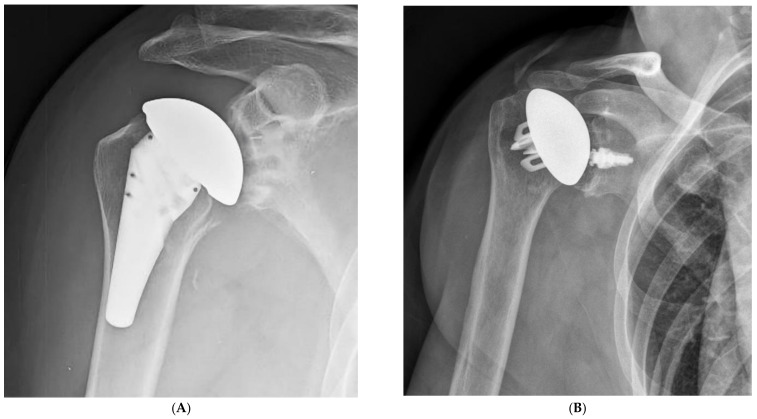
Ultrashort stems (**A**) and stemless prostheses (**B**) are at increased risk for malalignment.

**Figure 2 jcm-10-05151-f002:**
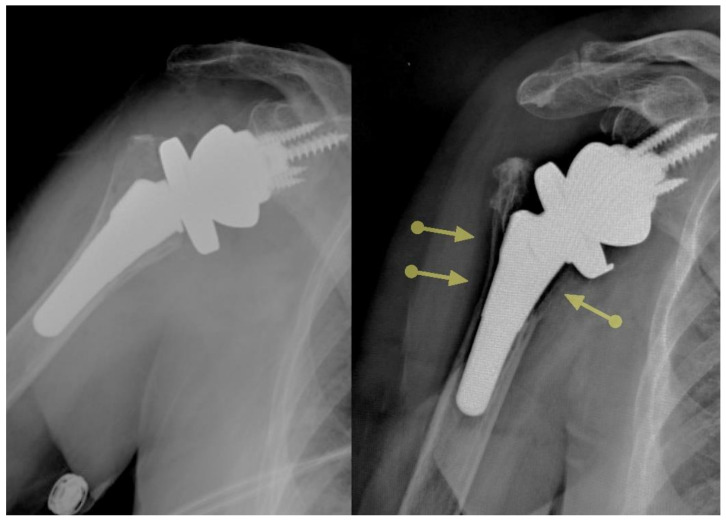
Certain ultrashort stems are associated with substantial stress shielding.

**Figure 3 jcm-10-05151-f003:**
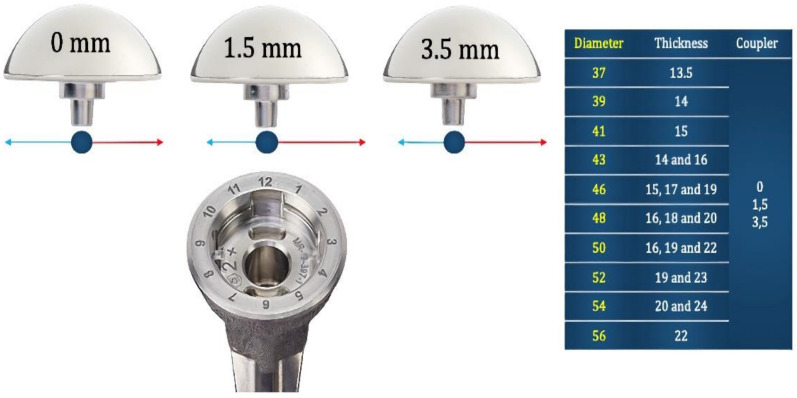
Options for replacement of the humeral head with one system for anatomic shoulder arthroplasty.

**Figure 4 jcm-10-05151-f004:**
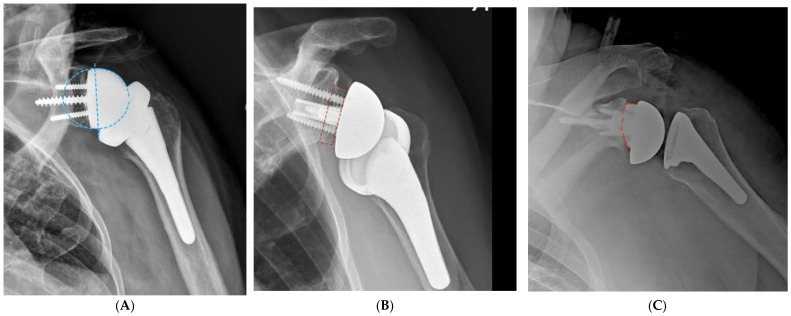
(**A**), Impingement-free range of motion is optimized with implantation of larger glenospheres with a lateral offset and an inferior overhang. Glenoid lateralization may be achieved with thicker glenospheres (**B**), the use of a bone graft under the baseplate (BIO-RSA), or the use of augmented baseplates (**C**).

**Figure 5 jcm-10-05151-f005:**
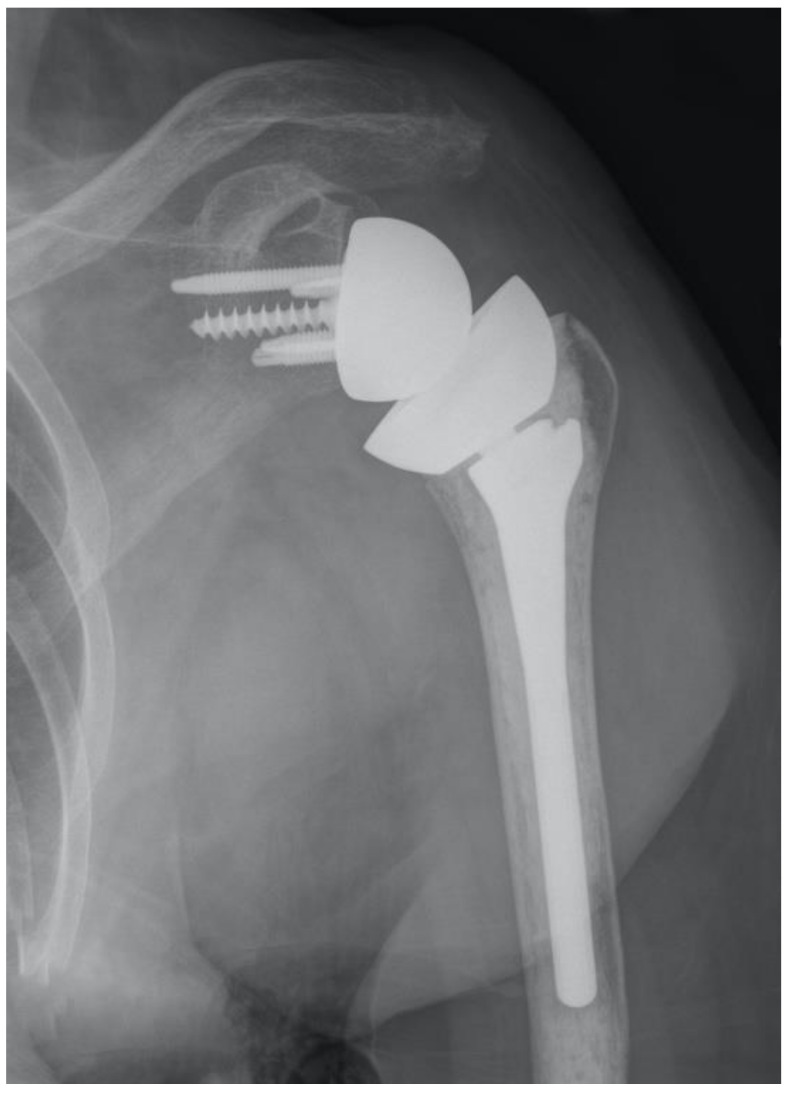
This implant was designed as an inlay, but its large size resulted in an onlay application most of the time.

**Figure 6 jcm-10-05151-f006:**
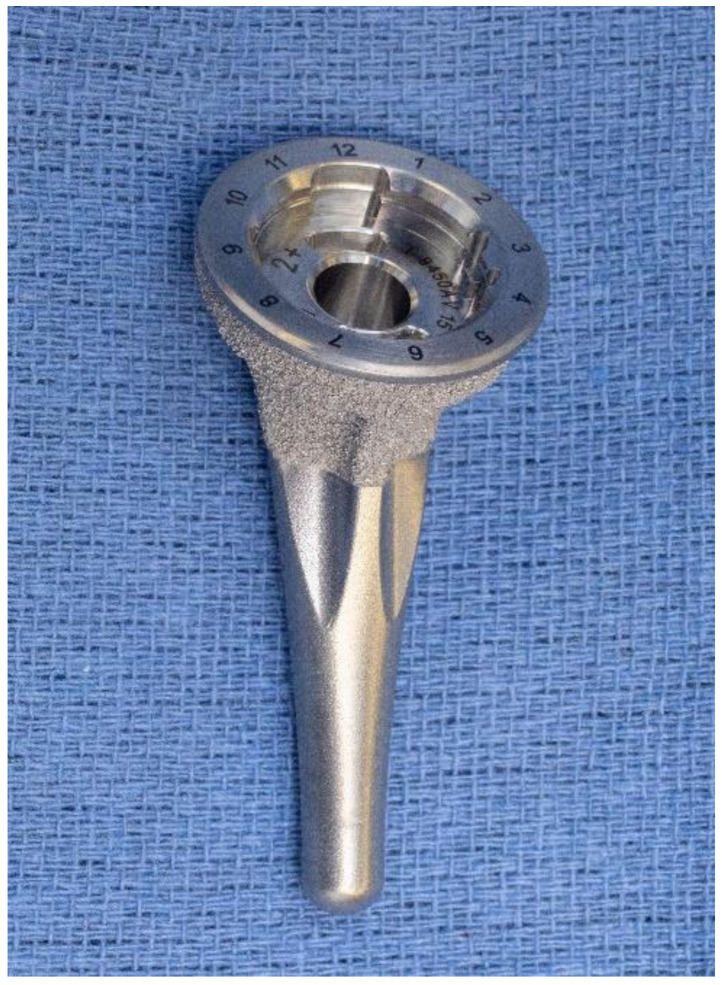
Contemporary platform of a short-stem humeral component designed for proximal fixation and loading.

**Figure 7 jcm-10-05151-f007:**
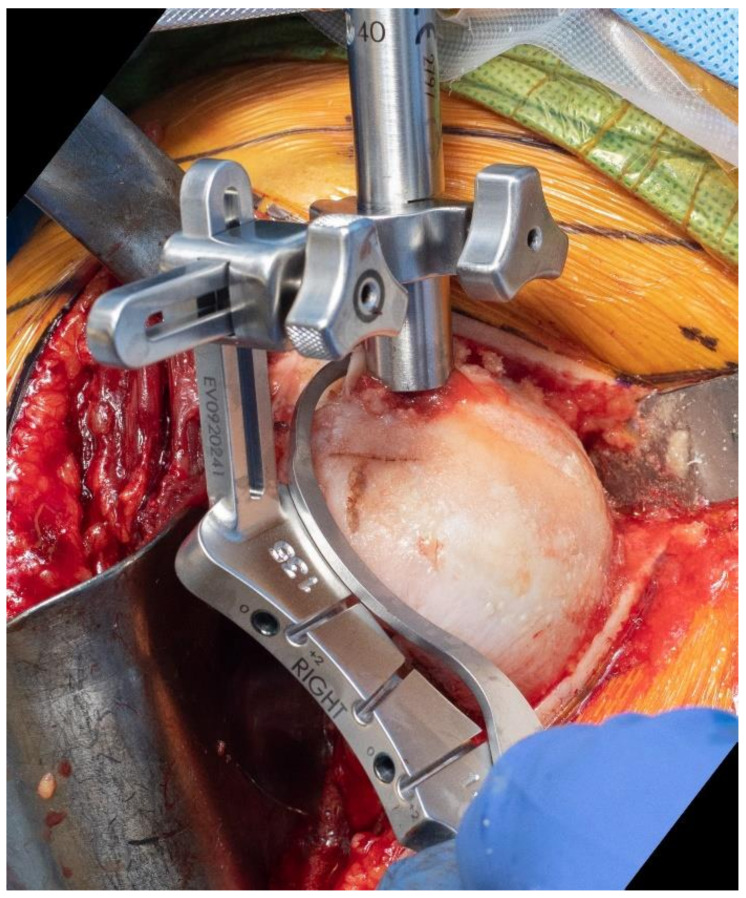
Use of an extramedullary cutting guide may facilitate predictable osteotomy of the humerus in a specific degree of inclination.

**Figure 8 jcm-10-05151-f008:**
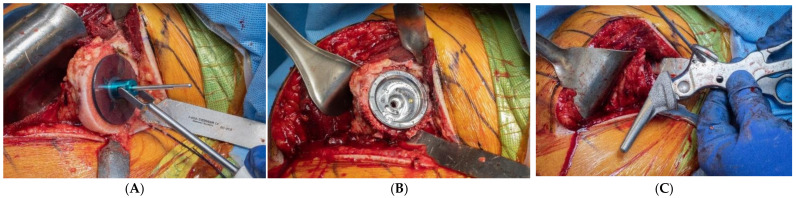
Primary component stability may be optimized by the implant fit at the periphery of the proximal humerus, within 2–4 mm of the cortical rim. (**A**), Sizing disk; (**B**), Trial; (**C**), Humeral component.

**Table 1 jcm-10-05151-t001:** Desirable features for contemporary humeral component design.

Platform: same component for anatomic and reverse arthroplasty Multiple anatomic head options for accurate restoration of humeral geometry Reverse configuration must accommodate large glenospheres with lateral offset and inferior overhang Proximal coating and proximal loading Peripheral metaphyseal fixation Short-stem and stemless offerings Streamlined instrumentation shared for stem and stemless Accurate execution of implant placement ▪ Preoperative planning software ▪ Cutting guides ▪ Patient-specific guides ▪ Navigation ▪ Robotics

**Table 2 jcm-10-05151-t002:** Intraoperative assessment of the soft tissue balance in anatomic shoulder arthroplasty.

Passive posterior translation of the humeral head in reference to the glenoid component of approximately 50% with spontaneous relocation (arm at 30 degrees of external rotation) Subscapularis can be repaired without excessive tension Satisfactory passive elevation, external rotation, and internal rotation The humeral head “spins” on the glenoid component surface in rotation without excessive translation

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
