# Peer review of "Current Concepts in Humeral Component Design for Anatomic and Reverse Shoulder Arthroplasty"

_jcm, 2021, doi:10.3390/jcm10215151_

Round 1

Reviewer 1 Report

Congratulations for your brief but very complete report about the humeral component in anatomic and reverse total shoulder arthroplasty. The evolution of the different systems is displayed accurately. The reader understands the role and the perspective of different humeral implant concepts.

As pointed out in the article, the role of lateralization of the humeral component remains unclear until now. This is probably one of the most interesting questions for the future.

I do not know if revisions should be included in this report…If so, it may be mentioned that there is still a lack of humeral revision implant with platform long stems that may replace the proximal humerus and the proximal metaphysis.

Maybe a word on cemented humeral components in primary shoulder arthroplasty could help the reader to understand if cement should be avoided in order to facilitate revisions.

Author Response

Dear Sir, thank you very much for your review of our manuscript. As pointed out in your review, our submission does not include humeral components used in the revision setting.

According to your suggestions, we have added a sentence to the manuscript stating that one of the drawbacks of cemented fixation of humeral components in the primary setting is the difficulties associated with component removal if that necessary in the future.

Reviewer 2 Report

This is an excellent summary of humeral prosthesis design for shoulder replacement.

My only suggestion would be to further discuss the effects of humeral lateralisation on RSA. I think this is an important topic at present. There are also designs that have the option of lateralisation within the humeral liner, that could be mentioned.

Not much on the version of humeral component in RSA and whether this should be standardised, patient-specific considering scapula position or combined retroversion with glenoid- although this may fall outside the scope of the paper.

Author Response

Dear Sir:

Thank you very much for your thoughtful review of our submission. We are glad to learn that for the most part you found it interesting. We have added some additional discussion on the effects of humeral lateralization. Regarding placement of the humeral component in various degrees of version depending scapular position in space and the orientation of the glenoid component, we agree with your observation that it exceeds the scope of this manuscript.